# High tropospheric ozone in Lhasa within the Asian summer monsoon anticyclone in 2013: influence of convective transport and stratospheric intrusions

Dan Li[1,2], Bärbel Vogel[1], Rolf Müller[1], Jianchun Bian[2,3], Gebhard Günther[1], Qian Li[2], Jinqiang Zhang[2,3], Zhixuan Bai[2], Holger Vömel[4], and Martin Riese[1]

[1]Institute of Energy and Climate Research: Stratosphere (IEK-7), Forschungszentrum Jülich, Jülich, Germany
[2]Key Laboratory of Middle Atmosphere and Global Environment Observation (LAGEO), Institute of Atmospheric Physics, Chinese Academy of Sciences, Beijing, China
[3]College of Earth Science, University of Chinese Academy of Sciences, Beijing, China
[4]Earth Observing Laboratory, National Center for Atmospheric Research, Boulder, CO, USA

**Correspondence:** Dan Li (da.li@fz-juelich.de) and Jianchun Bian (bjc@mail.iap.ac.cn)

**Abstract.** Balloon-borne measurements of ozone in Lhasa ($29.66°$ N, $91.14°$ E, $3650$ m above sea level) in August 2013 are investigated using backward trajectory calculations performed with the Chemical Lagrangian Model of the Stratosphere (CLaMS). Measurements show three time periods characterized by high ozone mixing ratios in the troposphere on 8, 11, and 18–20 August 2013 during the Asian summer monsoon (ASM) season. Here, we verified two different sources for the enhanced ozone values in the troposphere. First, transport of polluted air from the boundary layer, and second downward transport from the stratosphere by stratospheric intrusions. Air pollution from South Asia through convective and long-range transport plays a key role in enhancing middle tropospheric ozone mixing ratios up to 90% on 8 August and up to 125% on 11 August 2013 compared to monthly mean ozone of August 2013. Stratospheric air intruded from the northern high-latitudes to the southeastern flank of the ASM anticyclone to the troposphere and is identified as source of enhanced ozone according to backward trajectory calculation and satellite measurements by the Ozone Monitoring Instrument (OMI) and the Atmospheric Infrared Sounder (AIRS). Air parcels with high ozone moved from the high latitude lower stratosphere to the middle and upper troposphere. These air parcels are then transported to Lhasa over long-distances and enhanced upper and middle tropospheric ozone over Lhasa during 18–20 August 2013. Our findings demonstrate that the strong variability of ozone within the ASM anticyclone in the free troposphere is caused by transport from very different regions of the atmosphere.

*Copyright statement.* TEXT

## 1 Introduction

In the troposphere, ozone acts as an important greenhouse gas, which has a positive radiative forcing ($0.4 \pm 0.2\,\mathrm{W\,m^{-2}}$) through the direct or indirect greenhouse effect for the period 1750–2011. Although relatively short lived, it is therefore very

important for the radiation balance of the Earth's atmosphere (Myhre et al., 2013). Enhanced tropospheric ozone has an impact on radiative forcing of climate (Stevenson et al., 2013). Further small changes of ozone in the upper troposphere have an impact on surface climate (Riese et al., 2012). Thus, tropospheric ozone plays a major role in regional energy balance and in climate change. The variation of tropospheric ozone mixing ratios is associated with: (1) downward transport from the stratosphere, or upward transport from the boundary layer; (2) photochemical production induced by solar radiation and other chemical reactions involving lighting produced $NO_x$ or ozone-precursors from biomass burning and anthropogenic pollution. Stratosphere−troposphere exchange is affecting atmospheric tracer concentrations in a significant way via upward transport from the troposphere to the stratosphere and vice versa (Holton et al., 1995; Stohl et al., 2003; Fadnavis et al., 2010, 2018; Hoor et al., 2010; Gettelman et al., 2011; Pan et al., 2016). Biomass burning emissions of ozone precursors may cause enhance events of ozone in the free troposphere through photochemical production during tropospheric transport (Kita et al., 2002; Anderson et al., 2016) Stratospheric intrusions transport ozone-rich air downward into the troposphere (Eisele et al., 1999; Vogel et al., 2011; Weigel et al., 2012; Langford et al., 2015; Tao et al., 2018).

The Asian summer monsoon (ASM) anticyclone is the most intense circulation pattern in the Northern Hemisphere in the upper troposphere and lower stratosphere (UTLS) during boreal summer (Mason and Anderson, 1963) which is forced by deep convection over South Asia (Hoskins and Rodwell, 1995). Intense monsoon convection can transport tropospheric tracers (such as hydrogen cyanide (HCN) produced by biomass burning, carbon monoxide (CO), nitrogen oxides ($NO_x$) or aerosols) from the lower troposphere into the UTLS of the ASM anticyclone or its edge (Chen et al., 2012; Vogel et al., 2015; Tissier and Legras, 2016; Li et al., 2017; Fan et al., 2017b; Fadnavis et al., 2018). Because of the strong dynamical confinement of the ASM anticyclone in the UTLS region in summer (Ploeger et al., 2015; Pan et al., 2016; Vogel et al., 2016, 2018), tropospheric trace gases show a local maximum near the tropopause layer within the ASM anticyclone according to satellite measurements (e.g., Park et al., 2009; Randel et al., 2010; Vernier et al., 2015; Yan and Bian, 2015; Yu et al., 2017). At the edge of the ASM anticyclone, tropospheric tracers within the ASM anticyclone are transported outside, will affect trace gas concentrations in the UTLS result in significant changes in radiative forcing (Garny and Randel, 2016).

The ASM anticyclone is an active region for both troposphere-to-stratosphere and stratosphere-to-troposphere transport (e.g., Garny and Randel, 2016; Fan et al., 2017a), particularly the Tibetan Plateau region (Škerlak et al., 2014). In-situ balloon measurements in August 2015 in Kunming, China combined with satellite data and model simulations show that anthropogenic emissions from Asia enter to the upper troposphere and lower stratosphere and play a significant role in the tropopause aerosol layer formation within the ASM anticyclone (Yu et al., 2017). Vogel et al. (2016) show that the northeastern flank of the ASM anticyclone is a region where air masses from the ASM anticyclone are separated from the anticyclone and are subsequently transported into the extra-tropical lower stratosphere. When air parcels enter into the stratosphere, they have the potential to impact the regional climate in Asia (Vernier et al., 2015; Gu et al., 2016). Ozone concentrations in the planetary boundary layer over the Tibetan Plateau are most likely affected by intense deep stratospheric intrusion (Škerlak et al., 2014). Balloon measurements over the central Himalayas have shown that stratospheric intrusions enhanced ozone concentrations in the middle and upper troposphere (Ojha et al., 2014, 2017). Stratospheric intrusions occur at the northeastern flank of the anticyclone and transport dry ozone-rich air into the troposphere over northern India. Stratospheric intrusions or tropopause folds have in

addition the potential to influence surface weather including monsoon deficit rainfall (Fadnavis and Chattopadhyay, 2017). The upper tropospheric subtropical jets steam occur from eastern Asia to the mid-Pacific with high frequency (Koch et al., 2006). Stratosphere-to-troposphere transport along the subtropical jet steam occur over the Pacific Ocean. This is an important process for increasing ozone in the middle and upper troposphere in the region of the ASM (Trickl et al., 2011).

In particular, the Tibetan Plateau is a hotspot region for the two-way exchange between the stratosphere and troposphere (Škerlak et al., 2014). However, in situ measurements over this region of chemical compositions in the upper troposphere and lower stratosphere are limited (e.g., Bian et al., 2012; Li et al., 2017). Because of the sparse in situ observations over the Tibetan Plateau, there is a need for further in situ observations in this region (e.g., balloon or super-pressure balloon measurements) to obtain new insights into transport and exchange processes in this region and for climatological survey.

Balloon-borne measurements provided highly accurate water vapour and ozone profiles. Measurements with such balloon payloads have been carried out in Lhasa and Kunming, China (Bian et al., 2012; Li et al., 2017), Nainital, India, and Dhulikhel, Nepal (Ojha et al., 2014, 2017; Brunamonti et al., 2018), and in southern India (Vernier et al., 2018). Low ozone values measured in the upper troposphere in Lhasa were present by Li et al. (2017), rapid vertical transport associated with typhoon convection lead to this phenomenon. The ozone profiles in Li et al. (2017) also show anomalies of high ozone values in the

middle and upper troposphere over the Tibetan Plateau, however, they gave limited explanation about this phenomenon. In this paper, the detailed transport pathways of enhanced ozone values measured in August 2013 over Lhasa are analyzed. It is important to investigate the ozone variation over the Tibet Plateau, in order to quantify the uncertainty of the radiative forcing from tropospheric ozone in climate model. In this study, we combined these in-situ measurements with satellite data and trajectory calculations using the Chemical Lagrangian Model of the Stratosphere (CLaMS) model (McKenna et al., 2002;

Pommrich et al., 2014) to analyse the origin of high ozone structures found in the middle and upper troposphere in Lhasa over the Tibetan Plateau in August 2013. This paper is organized as follows: Sect. 2 describes the balloon sonde data, satellite data, and the CLaMS model. In Sect. 3, we present three case studies with enhanced tropospheric ozone in August 2013. A summary is given in the final section.

## 2   Measurements and trajectory calculations

### 25   2.1   Balloon-borne measurements

The SWOP (Sounding Water vapour, Ozone, and Particle) experiment was conducted in Lhasa (29.66° N, 91.14° E, 3650 m above sea level (a.s.l.)) in 2010, 2013, 2016, and 2018 and Kunming (25.01° N, 102.65° E, 1889 m a.s.l.) in 2009, 2011, 2012, 2014, 2015, and 2017 by Institute of Atmospheric Physics, Chinese Academy of Sciences during the summer monsoon period. The campaign was designed to collect the first long-term database of ozone, water vapour, and particles backscatter over the

Tibetan Plateau from surface to lower stratosphere with the aim to investigate and quantify the character of ozone and water vapour transport within the ASM anticyclone. Vertical profiles of ozone and water vapour shown in this study are from the SWOP campaign in August 2013. A total of 24 balloons were launched during nighttime between 22:00 BST (Beijing Standard Time, UTC+8) and 23:00 BST in Lhasa. The payload consists of an electrochemical concentration cell (ECC) ozonesonde

(Komhyr et al., 1995) to measure ozone, a cryogenic frost point hygrometer (CFH) (Vömel et al., 2007, 2016) to measure the frost (dew) point temperature for the temperature below (above) $-15°C$, and a compact optical backscatter aerosol detector (COBALD, developed at the Swiss Federal Institute of Technology, Zürich) backscatter sonde to detect aerosol or ice cloud backscatter (Brabec et al., 2012). An iMet radiosonde was used to transmit the CFH and Cobald data as well as to measure the

ambient temperature, pressure, relative humidity (RH), and wind speed/direction. Further details about the different balloon flights during the SWOP campaign in 2013 are given by Li et al. (2017).

The relative humidity over ice ($RH_i$) from CFH is defined as

$$RH_i = \frac{e}{e_{sat}} \times 100\% \tag{1}$$

where $e$ is the water vapour pressure calculated from the frost point or dew point temperature and $e_{sat}$ is the saturated

vapour pressure with respect to liquid water or ice, which is calculated from the ambient temperature using the Hyland$-$Wexler equation (Hyland and Wexler, 1983) for liquid water and Goff$-$Gratch equation (Goff and Gratch, 1946) for ice water. The $RH_i$ uncertainty is 5% in the tropopause layer (Vömel et al., 2016).

The colour index (CI) (Rosen and Kjome, 1991) is defined as

$$Colour\ index = \frac{BSRred - 1}{BSRblue - 1} \tag{2}$$

where BSRblue and BSRred are backscatter ratios at wavelengths of 455 nm and 940 nm, respectively. CI is used to separate in-cloud (CI>7) from clear sky (CI<7) (Vernier et al., 2015; Brunamonti et al., 2018).

## 2.2   Satellite data

The ozone monitoring instrument (OMI) is a nadir viewing near ultraviolet/visible charge$-$coupled spectrometer aboard the National Aeronautics and Space Administration's (NASA's) Earth observing system's Aura satellite (Levelt et al., 2006). In

this study we use the TOMS$-$Like total column ozone level-3 product (OMTO3e) with horizontal resolution 0.25×0.25. The TOMS version 8 algorithm is used to extract the vertical column ozone data using only two wavelengths (317.5 and 331.2 nm). The strong ozone absorption at 317.5 nm is used to derive total ozone, and the weaker absorb at 331.2 nm is used to estimate the effective surface reflectivity. The relative uncertainty on OMI-TOMS product is less than 5%.

The atmospheric infrared sounder (AIRS) on NASA's Aqua satellite is on a sun synchronous polar orbit. The instrument

employs a cross-track scanning hyper-spectral infrared spectrometer with 2378 spectral channels (Aumann et al., 2003). It is designed to provide twice daily global data sets for different constituents and temperature. Here we use the AIRS Level-2 ozone and water retrieval product version 6.0 (Olsen et al., 2013).

CloudSat is designed to probe the vertical structure of clouds and precipitation using a cloud profiling radar (CPR), as a component of the A-Train (Marchand et al., 2008). The CloudSat operational 2B Geometric Profile (2B-GEOPROF) data

product (Version R04) is used with 480 m vertical resolution. The vertical distribution of radar reflectivity is used to mark cloud layer. The echo mask values are greater than 20 dBZe indicating a false detection value below 16%.

## 2.3 Model

Fifty-day backward trajectories are started along the ascent profile of each balloon flight in Lhasa in August 2013. The diabatic trajectories are calculated using the CLaMS trajectory model (McKenna et al., 2002; Konopka et al., 2004; Pommrich et al., 2014). The trajectories of air parcels are calculated using the classical fourth-order Runge−Kutta method with a 1800-second time step. The CLaMS model employs a hybrid pressure-potential temperature coordinate, detailed information about the design of the vertical coordinate was presented by Konopka et al. (2007). The trajectory has been used to focus on the transport process within or around the ASM anticyclone (Vogel et al., 2015, 2018; Ploeger et al., 2015). Dynamic fields from the European Centre for Medium-range Weather Forecasts (ECMWF) interim reanalysis (ERA-Interim) (Dee et al., 2011) are used to drive the CLaMS model. The input dynamic fields are recorded every 6 hours on a $1° \times 1°$ horizontal grid with 60 hybrid vertical levels from the surface to 0.1 hPa. The vertical velocity on hybrid level is calculated using the diabatic heating budget including the cloud radiation, latent heat release, and mixing and diffusion (Ploeger et al., 2010). The trajectory model setup is the same as in Li et al. (2017).

## 3 Results

The ozone profiles over Lhasa show a pronounced daily variation of ozone mixing ratios (OMR) between 340 K and 420 K from 4 to 27 August 2013 (Fig. 1). The lapse-rate tropopause is calculated from measured temperature profiles using the lapse-rate criteria of the World Meteorological Organization (WMO) definition (WMO, 1957). The lapse rate tropopause in Lhasa range from 365 K to 410 K in August 2013. Low ozone values were measured in the upper troposphere on 11, 19, and 24 August 2013 (Fig. 1). Li et al. (2017) have combined balloon-borne measurements with CLaMS trajectory model to highlight the low ozone structures in the upper troposphere over Lhasa. A major reason why ozone mixing ratios in the upper troposphere remain low is the impact from tropical cyclones, which transport marine boundary layer air with low ozone to the upper troposphere over Lhasa. However, two profiles with extreme high ozone mixing ratios, up to 180 ppbv are found in upper troposphere (355–365 K) on 8 August and in the troposphere from 330 K to 352 K on 11 August 2013 (Fig. 1). Further, episodes with high ozone mixing ratios occurred from the middle troposphere to the lower stratosphere during 18–20 August 2013. Whereas Li et al. (2017) discuss low ozone values in August 2013, here we will focus on extreme high ozone values.

### 3.1 High ozone and long-range transport on 8 August 2013

Figure 2a shows the vertical profiles of temperature, OMR, monthly mean ozone, RH, and OMR relative change. The positive OMR anomalies with value up to 180 ppbv occur between 355 K and 365 K, which is below the lapse rate tropopause (378 K) at 22:41 BST on 8 August 2013 (Fig. 2a left). The OMR relative change shown in Fig. 2a (right) shows the percentage deviation of the ozone profile observed on 8 August from the monthly mean ozone profile obtained by averaging 24 profiles over Lhasa in August 2013. The data shows ozone anomalies up to 90 % on August 8. The high ozone value is marked by dark shade for the OMR relative changes above 30%; in the following, we solely focus on these air masses. Unfortunately, $RH_i$ from CFH are

not available above 335 K. Instead, the relative humidity (RH) from iMet was used, with useful data just below 350 K (above $-40°C$). Below $-40°C$, RH will not be used, because of the detection limit of the radiosonde humidity sensor.

The potential vorticity (PV) along the 30-day backward trajectories of air parcels within high ozone layer between 355 K and 363 K was shown in Fig. 2b as a function of time and potential temperature. According to their different pathways, backward

trajectories of air parcels could be divided into two clusters. Both of them experienced strong uplift processes with potential temperature increasing from 330 K to 360 K on 20 and 29 July. The tracks of 20-day backward trajectories of air parcels around 355–363 K isentropes initialized on 8 August 2013 are shown colour-coded by date in Fig. 2c. Air parcels started their ascent near the Himalayas and were uplifted to 360 K isentrope from 19 to 21 July and then moved horizontally to East Asia following the ASM anticyclone. Finally, the air parcels moved westerly around the anticyclone circulation before they arrived in Lhasa

on 8 August 2013. The transport time for the air parcels from the lower troposphere of the Himalayas to the upper troposphere over Lhasa is less than 20 days for the whole pathway.

Figure 2d gives the boundary layer geolocation of air parcels (the same as in Fig. 2b and c), where they experience strong uplift through convection. The uplift rate of air parcel is defined as $(\theta_{t+\delta t} - \theta_t)/\delta t$. When the uplift rate is greater than $9\,\mathrm{K\,day}^{-1}$ (an empirical value), the strong uplift process of air parcels will be recognized. The point that air parcels start ascent is marked

as the geolocation. We find that most of air parcels were from the Himalayas, where strong uplift occurred frequently. South Asia, the area adjacent to the Himalayas is usually a strong source region of air pollution, which is caused by natural (e.g., biomass burning) and anthropogenic processes (Cong et al., 2015). In addition, ozone is photochemically enhanced by reactions involving ozone precursors from biomass burning. After reaching the upper troposphere, the polluted air masses that were transported over long distances, made the best possible contribution to high tropospheric ozone over Lhasa at nighttime on 8

August 2013.

## 3.2  High ozone and high water vapour on 11 August 2013

The vertical variability of OMR, monthly mean ozone, temperature, the OMR relative change, $\mathrm{RH_i}$, and colour index on 11 August 2013 is shown in Fig. 3a. Positive ozone anomalies (Fig. 3a left) and the OMR positive variance (Fig. 3a right) appeared from the surface to the mid-troposphere (330–353 K) at 22:40 BST on 11 August 2013. The $\mathrm{RH_i}$ shows high value ($>70\%$)

between 336 K and 350 K. Colour index from the COBALD shows that a thick cirrus cloud layer occurred below 349 K. The finding of high ozone concentration accompanied by high water vapour values within a thick cloud layer is unusual high over the Tibetan Plateau in our measurement.

PV values along the 50-day backward trajectories of air parcels within high ozone mixing ratios between 340 K and 354 K are shown in Fig. 3b. The results show that the PV values are less than 1.5 PVU in the troposphere. Air parcels were uplifted to

the upper troposphere through convection, then they are transported over a long distance and descended before they arrived in Lhasa. Three-dimensional backward trajectories of air parcels within the high ozone structure were shown in Fig. 3c. Particles were first uplifted to the upper troposphere and these were moved around the ASM anticyclone within 4–5 weeks. Air pollution from South Asia boundary layer has the potential to impact the ozone structure over Lhasa via transportation around the ASM anticyclone within 4–5 weeks.

Fig. 3d gives the boundary layer geolocation, where parcels experienced strong uplift. Most of the air parcels are transported from South/Southeast Asia, a region with high air pollution to Lhasa. Few of the air parcels originated from South China and the Tibetan Plateau. Obviously, the $RH_i$ is controlled by local microphysical cloud processes according to the colour index (CI>7, in-cloud). The horizontal scale of the cloud is less than $1 \times 1$ degree according to the MODIS satellite (figure not shown). Air pollutants from South Asia or Lhasa may contribute to the positive ozone anomalies over Lhasa through photochemical production, however, due to the detection limit for chemical constituents that produces ozone over the Tibetan Plateau during this period, it is not clear what are the reasons for high ozone within cirrus clouds.

## 3.3 Stratospheric intrusion on 18−20 August 2013

Figure 4a shows the total column ozone (TCO) from the OMI satellite along with the geopotential height, PV at 150 hPa, and the sea-level pressure of typhoon Utor on 16 August 2013. The PV clearly displays a filament structure extending from northeastern Asia to the southeastern edge of the ASM anticyclone on 16 August 2013. TCO displays the same structure as the PV over northeastern Asia, where stratospheric intrusions occur frequently (Li and Bian, 2015; Song et al., 2016). The enhanced TCO values are first of all indicative for an extra-tropical intrusion. Since the stratosphere contributes most to the column, it is also indicative for an intrusion of stratospheric air. This is further supported by the good correlation of TCO with PV values at 150 hPa. The PV on 370 K isentropic surface shows the same structure as PV at 150 hPa pressure level (Fig. 4b). The stratospheric intrusion with high ozone mixing ratios is also reflected in low water vapour values observed by AIRS at 83.666 hPa during its ascending track on 16 August 2013 (Fig. 4c and 4d). The filament structure is subsequently separated from the stratosphere. The remnant of the stratospheric intrusion (Rossby wave breaking filament) moves westward along the easterly wind flow at the southern flank of the ASM anticyclone. During the next few days, this broken filament structure moves westward about 3,000 km. It arrives in Lhasa and is captured by the ozonesondes launched on 18, 19 August 2013 (Fig. 4e−f), and on 20 August 2013 (figure not shown), contributing to the positive ozone value measured in the UTLS region over Lhasa.

Figure 5a shows the vertical variability of ozone, monthly mean ozone, temperature, $RH_i$, colour index, and the OMR relative change for 18 August 2013. The positive OMR relative change appears in the troposphere around 350 K and 363–373 K, and the lower stratosphere (Fig. 5a right). Two characteristic minima of OMR relative change occur between 352–357 K and 375–390 K on 18 August. $RH_i$ is negative correlated with OMR anomalies on these isentropic surfaces. $RH_i$ near the tropopause (384 K) is greater than 100%. Indeed, an ice cloud layer was observed near the tropopause layer according to the colour index (CI>7) from COBALD backscatter measurements. Supersaturation is observed within the ice cloud.

In order to investigate in detail the variance of ozone profiles measured on 18 August 2013, the PV along the 50-day backward trajectories from the CLaMS model is displayed in Figure 5b. Parcels in the upper troposphere (363–373 K) originate from the dry stratospheric intrusion layer. There is evidence for mixing processes that occurred between air parcels with high PV from stratosphere and low PV value from the troposphere, while air parcels in the middle troposphere (around 350 K) with high ozone and low water vapour originate from the thin intrusion layer. Air parcels around 350 K experienced a weak uplift during 13–14 August 2013. Figure 5c shows the tracks of backward trajectories of air parcels on 350–352 K and 363–373 K on 18 August,

respectively. Only 20-day backward trajectories of air parcels are shown colour-coded with potential temperature here. This is long enough to show the intrusion pathway. The thin intrusion layer with low potential temperature (338 K) moved toward the north around the anticyclone and then shows a strong equatorward movement from 60° N to 30° N around 60° E. The intrusion layer in the upper troposphere experienced mixing process according to the PV along the trajectories. Air parcels with high

PV values from the Northern Hemisphere were mixed with air parcels with low PV from the region at approximately 35° N, 60° W over the North Atlantic. Although ozone-rich air mixed with ozone-poor air from the equatorial region, the positive ozone anomalies still appeared in the upper troposphere on 18 August 2013. As Riese et al. (2012) show, ozone concentrations are sensitive to mixing strength in the lower stratosphere.

Positive ozone mixing ratio anomalies were also captured on 19 and 20 August 2013. The variability of ozone vertical

structure is significant in the middle troposphere on 19 August. Ozone and water vapour show strong anti-correlation below 355 K. The OMR relative change shows large increase in the tropopause region (368–408 K), up to 90% (Fig. 6a). The 50-day backward trajectories of air parcels in Fig. 6b indicate diabatic decent transport process, especially the air cluster in the middle troposphere (between 347 K and 355 K). The PV along the trajectories of air parcels between 368 K and 380 K display high PV with values greater than 6 PVU. Figure 6c shows the tracks of backward trajectories of air parcels within high ozone in

the middle and upper troposphere on 19 August. The stratospheric intrusion in the middle troposphere has the same transport pathway as on 18 August and also experienced an uplift process around 14 August. Air parcels near the tropopause layer (Fig. 6a) originated from the northern Hemisphere with high ozone and high PV. The equatorial regions contribute little to air parcels on 19 August compared to the case on 18 August. That is why the OMR relative change near the tropopause layer on 19 August is higher than OMR relative change on 18 August.

The ozone vertical structure and $RH_i$ also show an anti-correlation in the middle troposphere on 20 August. In the troposphere, the OMR relative change near 355 K is higher than 30%. The OMR relative change on 20 August shows minor increases in the tropopause region (Fig. 7a) and is also weaker than on 19 August. The height of lapse rate tropopause on 20 August is lower than on 18 and 19 August. Fig. 7b shows the PV along the backward trajectories of air parcels in the middle troposphere (355 K) on 20 August. The intrusion in the middle troposphere has the same transport pathway as the one on 18

August and also experienced an uplift process around 14 August (Fig. 7c).

In order to explore the intrusion structure in the middle troposphere, the ozone mixing ratio from AIRS satellite at 346 hPa is shown on 13 August 2013, 08:30 UTC with PV at 350 hPa from ERA-Interim on 13 August 2013, 06:00 UTC (Figure 8a). Both ozone and PV show that a filament structure extend from the south of Afghanistan via Tian Shan to Russia in the northern extra-tropics. Meanwhile, the latitude−pressure cross section of the AIRS ozone along 70° E displays a tropopause

fold. The intense stratospheric intrusion transported air with high ozone from the tropopause layer downward to the surface of the mountain at 40° N (Fig. 8b). The structure of the tropopause fold from PV and ozone are a little shifted, because the time difference between AIRS ascending time (around 08:30 UTC) and ERA-Interim reanalysis data (06:00 UTC).

Figure 9a−c show the cross section of the square of the Brunt−Vaisala frequency ($N^2$) along the average longitude of the bottom air cluster in Fig. 5b on 10 August at 06:00 UTC, on 12 August at 18:00 UTC, and on 13 August at 12:00 UTC with

potential temperature, PV, zonal wind, and the lapse rate tropopause calculated from the ERA-Interim reanalysis. The radar

reflectivities measured by the CloudSat's CPR are shown in Fig. 9d. Air parcels are located in the extra-tropical lowermost stratosphere (350−380 K) above the lapse rate tropopause (near 330 K) on 10 August 2013 at 06:00 UTC (Fig. 9a). These parcels moved equatorward and arrived at the poleward edge of westerly wind jet two days later, where a tropopause fold occurred (Fig. 9b). Parcels continue to move along the isentropic surfaces and cross the tropopause region from the polar lowermost stratosphere to the upper troposphere in the mid-latitude on 13 August 2013 at 12:00 UTC (Fig. 9c). Overall, it takes 2–3 days for air parcels to cross the tropopause. Both the pathway and timescale of transport are consistent with the analysis of other deep stratospheric intrusions that occurred over North America (Langford et al., 1996; Vogel et al., 2011; Kuang et al., 2012; Lin et al., 2015) or Europe (Stohl and Trickl, 1999; Trickl et al., 2010, 2014) associated with the polar jet stream. It is interesting that in our case, air parcels are affected by strong convection in the troposphere, after they are transported from the stratosphere downward into the upper troposphere. The strong convection lifted air parcels to high altitude, which can be seen from the CloudSat radar reflectivity (dBZe) (Fig. 9d). This uplift process can be seen clearly on 14 August 2013 in Fig. 5b. The extra-tropical tropopause is located between the upper troposphere and lower stratosphere and intersects the isentropic surface, which acts as a dynamic barrier for tracer transport (Gettelman et al., 2011). Ozone in the extratropics exhibit large gradients in the UTLS. There is a net downward transport of ozone from the stratosphere to the troposphere along the isentrope at the poleward edge of the jets (Yang et al., 2016). The tropopause fold transports air parcels quasi-isentropically from the lowermost stratosphere with high ozone mixing ratio to the troposphere within the ASM anticyclone, contributing to high tropospheric ozone over Tibetan Plateau on 18, 19, and 20 August 2013.

## 4 Discussion and conclusions

Balloon-borne measurements of ozone and water vapour mixing ratios over Lhasa in August 2013 are investigated using the OMI and AIRS satellite data and backward trajectory calculations using the CLaMS model. We focus on enhanced ozone mixing ratios observed in the middle and upper troposphere over Lhasa on 8, 11, and 18–20 August 2013.

For 8 and 11 August, trajectory calculations show that the enhanced ozone mixing ratios may result from air pollution originating from south Asia, which reached the measurement location through convective uplift and quasi-horizontal long-range transport within 20 days. Here we mainly focused on the 20-day backward trajectories. To study transport processes in the region of the Asian monsoon, a trajectory length between few days to few months is found in the literature (e.g., Chen et al., 2012; Bergman et al., 2013; Vogel et al., 2014; Vernier et al., 2015; Garny and Randel, 2016; Li et al., 2017). To study the convective events, a trajectory length of a few days to 20 days is sufficient. Garny and Randel (2016), using isentropic trajectory calculations, also show that air parcels originating at 360 K within anticyclone were confined within the ASM anticyclone for 10–20 days. Li et al. (2017) using the Lhasa's balloon data in 2013 and the CLaMS trajectory model, have shown that air parcels with low ozone from marine boundary layer over the western Pacific are the dominant source of low ozone in the tropopause layer in Lhasa resulting from very strong uplift (1–4 days) by typhoons and subsequent horizontal long-range transport (4–10 days) within the ASM anticyclone. Thus, 20-day backward trajectories from the CLaMS model for the middle and upper troposphere within the ASM anticyclone are generally appropriate. However, to analyze aged air masses transported in the

region of the ASM anticyclone or originating in the stratosphere trajectories longer than 20 days are necessary (e.g., Chen et al., 2012; Bergman et al., 2013; Vogel et al., 2014; Garny and Randel, 2016). The longer the trajectories run, the larger are the uncertainties due to trajectory dispersion and the lack of mixing processes (McKenna et al., 2002). For trajectories from the stratosphere we calculated a set of trajectories close to the location of the measurement (not shown here) and found that all these trajectories show a very similar transport pattern over the last 20 days.

During the ASM period, deep convective clouds occurred over the South Asian subcontinent, the southern Himalayan region and the Tibetan Plateau that caused lightning activity (Qie et al., 2014). $NO_x$ produced by lighting has the potential to impact upper tropospheric ozone via photochemical reactions (Kita et al., 2002; Schumann and Huntrieser, 2007; Fadnavis et al., 2015; Gottschaldt et al., 2018). Uplifted ozone precursors and $NO_x$ produced by lighting contribute to photochemical ozone production in the upper troposphere in the ASM anticyclone. An ozone increase by 30% is found over the monsoon convection region due to $NO_x$ enhanced by lightning according to sensitivity simulations with and without lightning (Fadnavis et al., 2015). In our case studies, ozone is increased by 90%, compared to monthly mean ozone, this value is still higher than the ozone increase due to lightning reported by Fadnavis et al. (2015). Lightning is an additional source of tropospheric ozone, which is not included in the trajectory calculations neglecting chemistry employed here. However, in addtion to lightning produced $NO_x$, ozone precursors likely uplifted by convection are necessary. Therefore, convective uplift is the basis transport mechanism for enhanced ozone in the middle and upper troposphere. Enhanced ozone measured over Lhasa is caused by uplift of ozone-rich air from the lower troposphere or produced by lightning, it is likely caused by a combination of both with a transport time from the planetary boundary layer to Lhasa of 20–30 days.

The ozone enhancement observed during the period 18–20 August 2013 resulted from a tropopause folding and a stratospheric intrusion. The northern side of the Tibetan Plateau is a hotspot region for transport processes from the stratosphere to the troposphere, which are most likely due to tropopause folding events (Sprenger et al., 2003; Škerlak et al., 2014). These findings are corroborated by our case studies. Our results show that the stratospheric air masses characterized by high ozone, which are transported downwards are subsequently transported quasi-horizontally over large special scales in the ASM anticyclone. The stratospheric intrusion occurred over the southeastern edge of the ASM anticyclone, transported air equatorwards with high ozone and low water vapour. Stratospheric intrusions contribute to high ozone and low water vapour values in the upper troposphere over Lhasa within the ASM anticyclone, thereby highlighting the important role of large-scale atmospheric dynamic transport for the trace gas budget in this region.

Satellite data and the trajectory calculations from the CLaMS model indicate that both the stratospheric intrusions and convective transport of air pollution play a major role in enhancing middle and upper tropospheric ozone over Lhasa, China. The PV values along the backward trajectories for the convective transport and the stratospheric intrusion are different and therefore a good indicator for the transport pathway of air masses. The PV values along the trajectories are less than 2 PVU for the convective transport case, and greater than 6 PVU for the stratospheric intrusion when air parcels originated in the extra-tropical lower stratosphere. The PV values decrease when air parcels cross the lapse rate tropopause from the lower stratosphere to the troposphere. Tropical cyclones, which can transport marine boundary layer air with low ozone to the upper troposphere, lead to low ozone values in the upper troposphere in Lhasa (Li et al., 2017). The ozone variability in the middle

and upper troposphere over Lhasa in 2013 is the result of different advective and convective transport processes transporting air parcels from different regions to Lhasa.

Our studies indicate that ozone variations in the UTLS over Lhasa are associated with a combination of processes, named by the intrusion of ozone-rich stratospheric air, the upward transport of low ozone concentrations in the marine boundary layer by tropical cyclones, and the uplift of polluted air rich in ozone in South Asia. Case studies are a first step to investigate and understand ozone variations within the ASM anticyclone; future work will focus on quantifying the long-term ozone and water variation in the UTLS within the ASM anticyclone, according to the 10-year balloon data from the SWOP campaign. Moreover, the SWOP campaigns will continue to be conducted for the next few years to extend the data set allowing climatological studies.

*Data availability.* ERA-Interim meteorological reanalysis data are free available from the web page: http://apps.ecmwf.int/datasets/data/interim-full-daily/. The OMI ozone data were download from https://search.earthdata.nasa.gov/. The AIRS Level−2 data used in this study can be obtained at https://airsl2.gesdisc.eosdis.nasa.gov/data/Aqua_AIRS_Level2/. The CloudSat products are provided on www.cloudsat.cira.colostate.edu. The SWOP data of this paper are available upon request to Jianchun Bian (bjc@mail.iap.ac.cn). The CLaMS backward trajectories calculations can be requested from Dan Li (lidan@mail.iap.ac.cn).

*Acknowledgements.* Ozone and water vapour data are from the SWOP campaign, which is funded by the National Natural Science Foundation of China (41605025, 41675040, and 91637104). Our activities contribute to the European Community's Seventh Framework Programme (FP7/2007-2013) as part of the StratoClim project (grant agreement no. 603557). This work was supported by International Postdoctoral Exchange Fellowship Program 2017 under grant no. 20171015 and China Postdoctoral Science Foundation (2015M581153). Finally, we wish to thank two anonymous reviewers for very constructive suggestions.

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

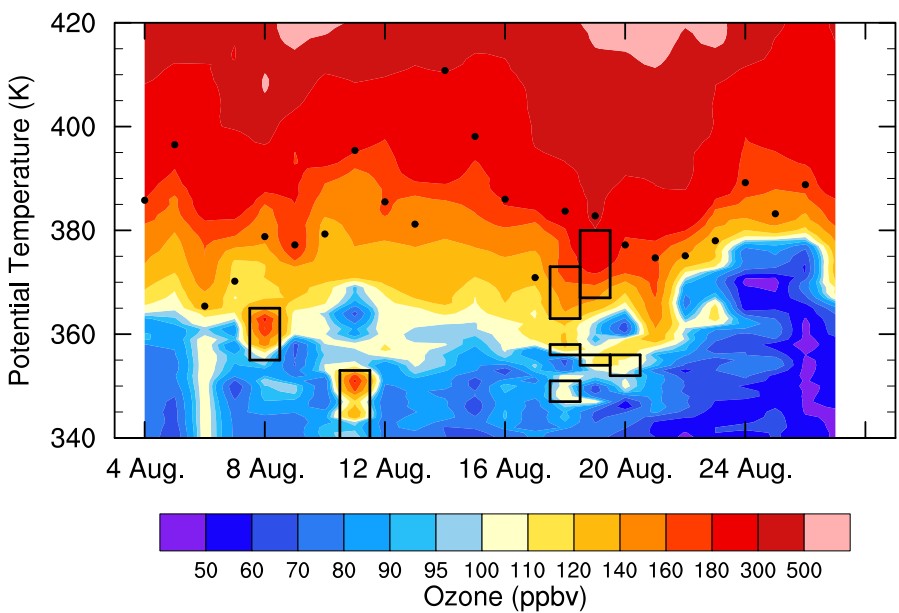

**Figure 1.** Daily variations of ozone mixing ratios (OMR) between 340 K and 420 K during 4–27 August 2013 from balloon measurements launched in Lhasa. The black dots represent the lapse rate tropopause. High tropospheric ozone is shown by the black rectangle.

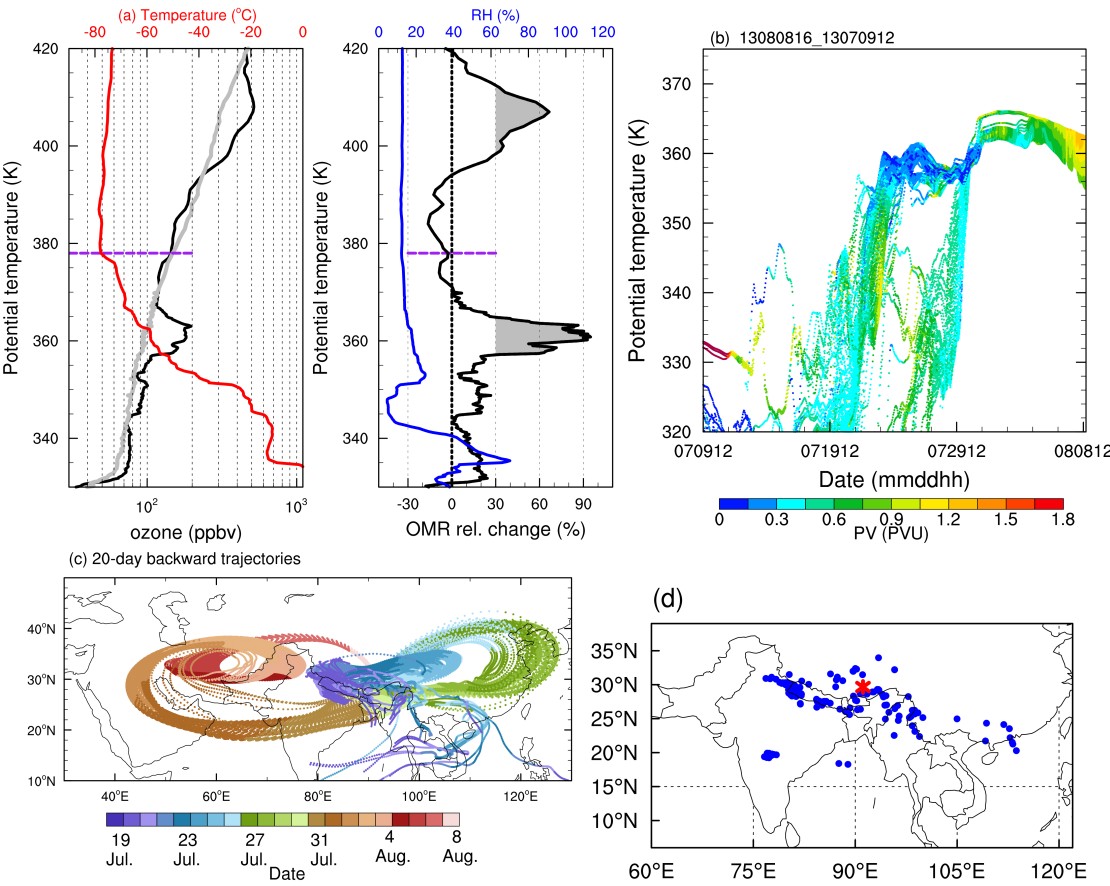

**Figure 2.** (a) Vertical profiles of ozone (black), monthly mean ozone (grey), and temperature (degree in red), OMR relative change, and relative humidity (blue) in Lhasa on 08 August 2013 at 14:00 UTC. The horizontal line marks the lapse rate tropopause. (b) Potential vorticity (1 PVU=$10^6$ K m$^2$ kg$^{-1}$ s$^{-1}$) along the 30-day backward trajectories of air parcels within the high ozone layer (355–362.3 K) as a function of time and potential temperature. (c) 20-day backward trajectories of air parcels between 355 K and 362.3 K at the Lhasa site on 08 August 2013 are shown colour-coded by date. (d) Geolocation of air parcels experienced strong uplift in the positive ozone mixing ratio anomalies. The red asterisk marks the location of Lhasa.

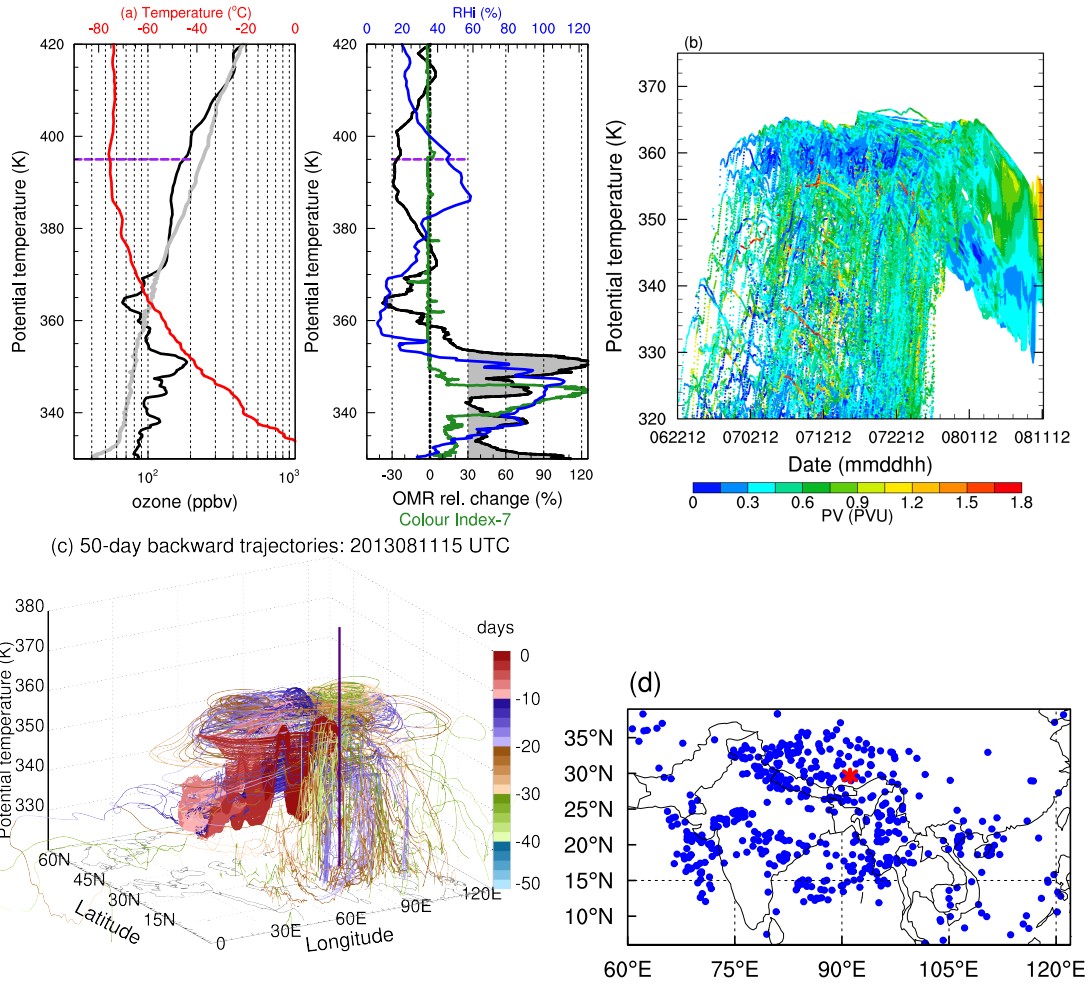

**Figure 3.** The same as Fig. 2 except for (c) but on 11 August 2013. (a) Colour index (CI) minus 7 (CI−7>0 marks the cirrus cloud layer, green line). (c) Three-dimensional backward trajectories colour-coded by days.

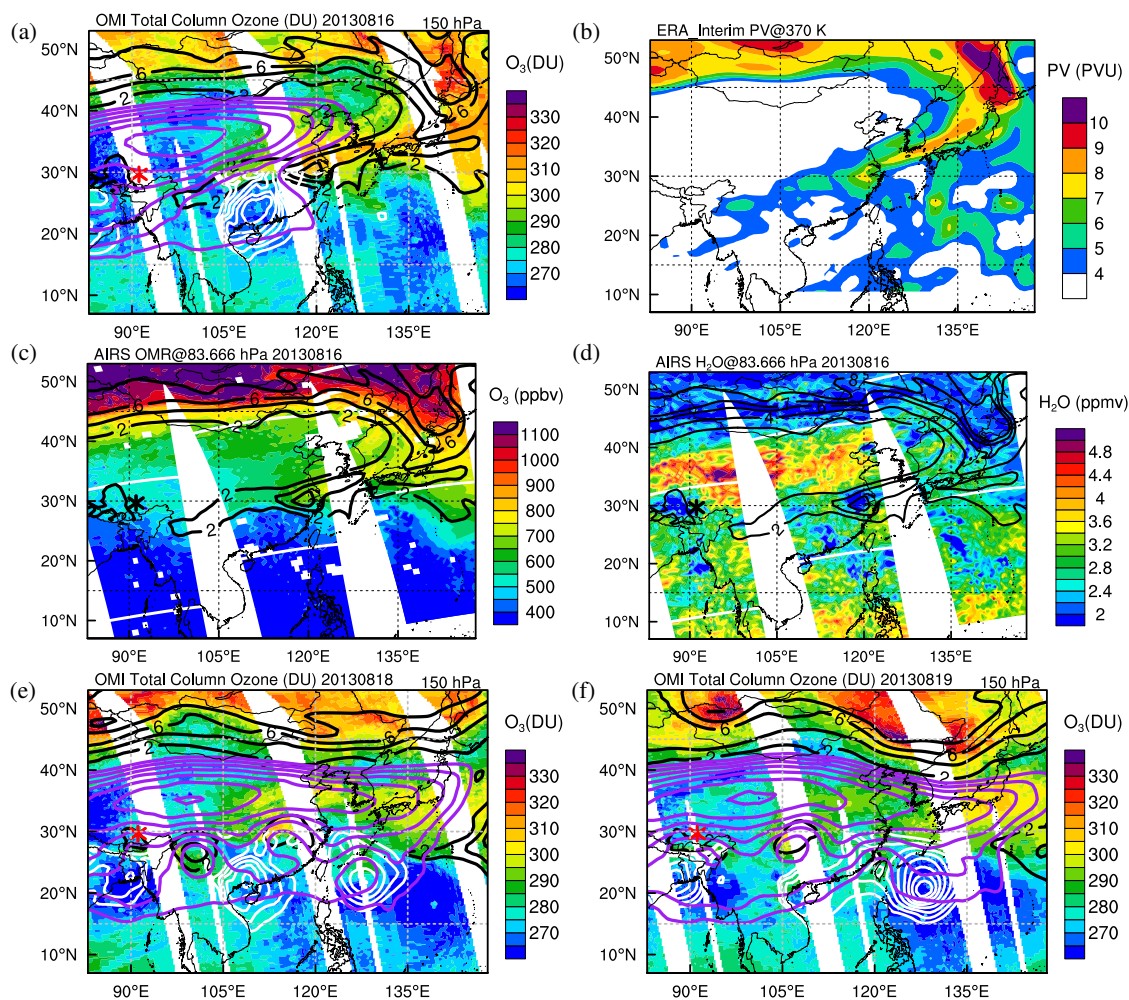

**Figure 4.** Total column ozone (DU) as measured by OMI with the geopotential height (purple line) and PV (black line, in PVU) at 150 hPa, and the sea-level pressure (white line) of tropical cyclone on (a) 16, (e) 18, and (f) 19 August 2013 using ERA-Interim data. (b) The PV (shade, PVU) on 370 K surface on 16 August 2013. (c) Ozone and (d) water vapour mixing ratios for 83.666 hPa from AIRS on 16 August 2013 with PV at 150 hPa from ERA-Interim. The asterisk marks the location of Lhasa.

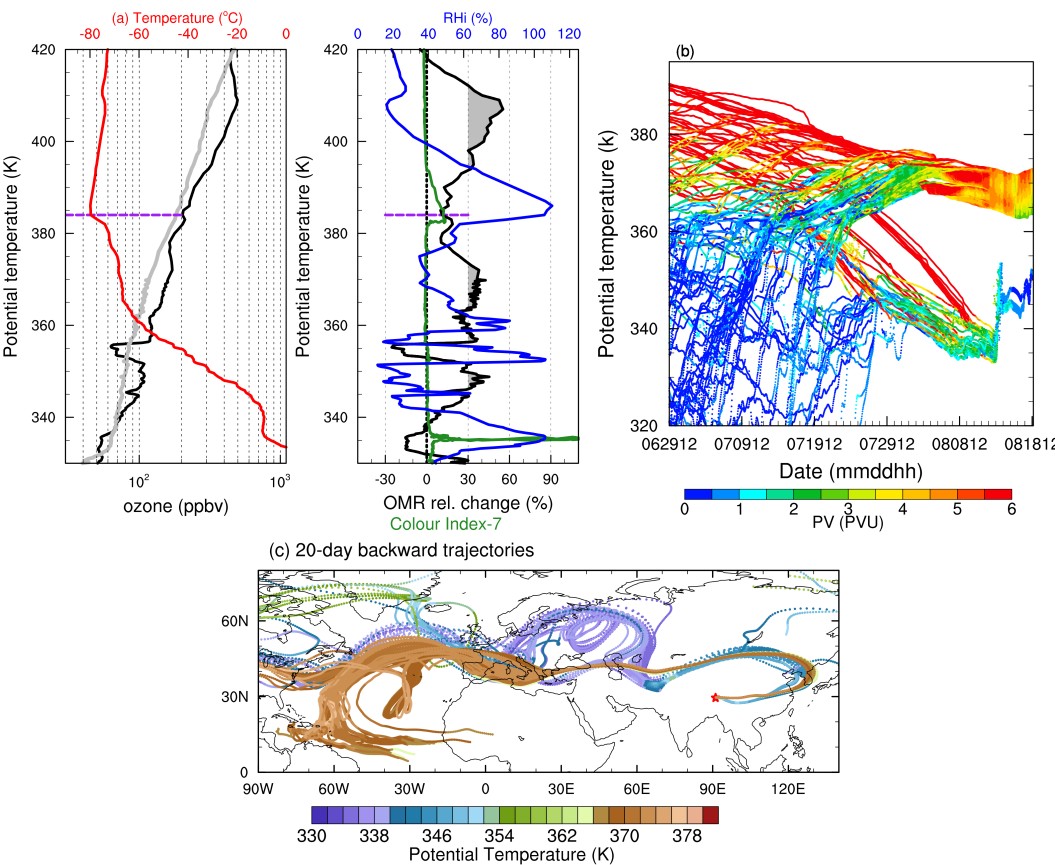

**Figure 5.** Panels (a, b) are the same as Fig. 3a−b but on 18 August 2013. (c) 20-day backward trajectories of air parcels within high ozone layer are shown colour-coded by potential temperature.

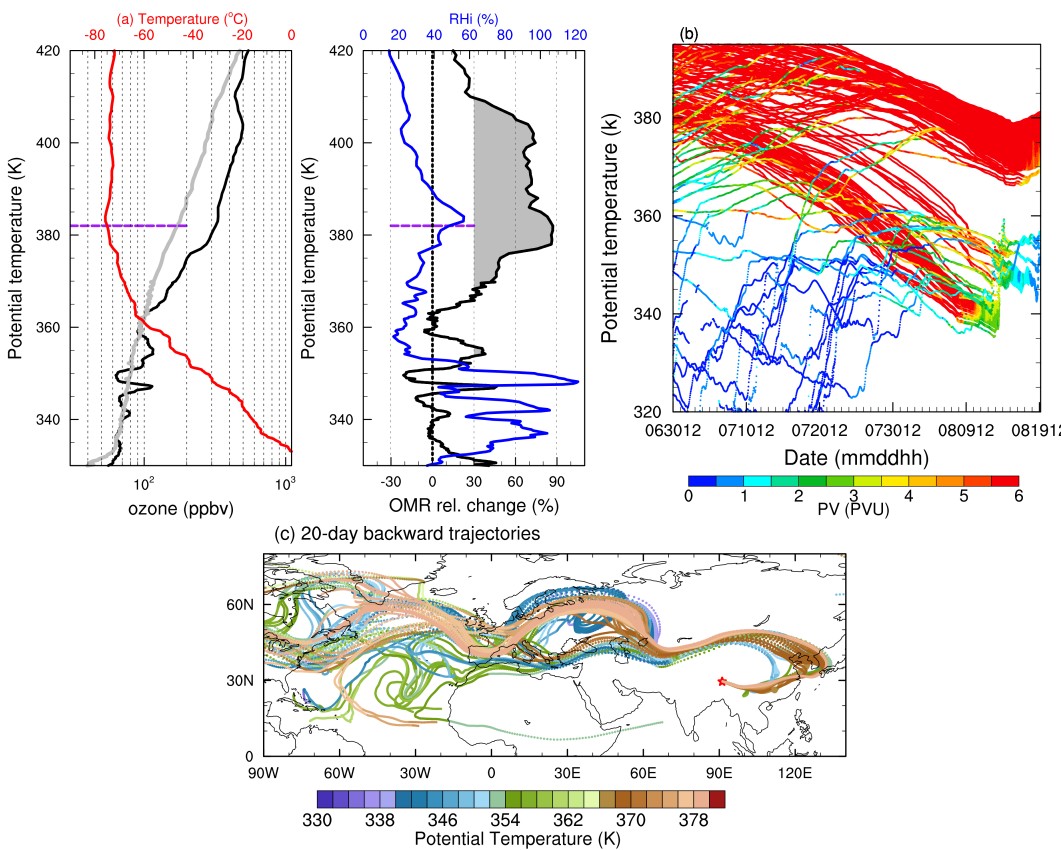

**Figure 6.** The same as Fig. 5, but on 19 August 2013.

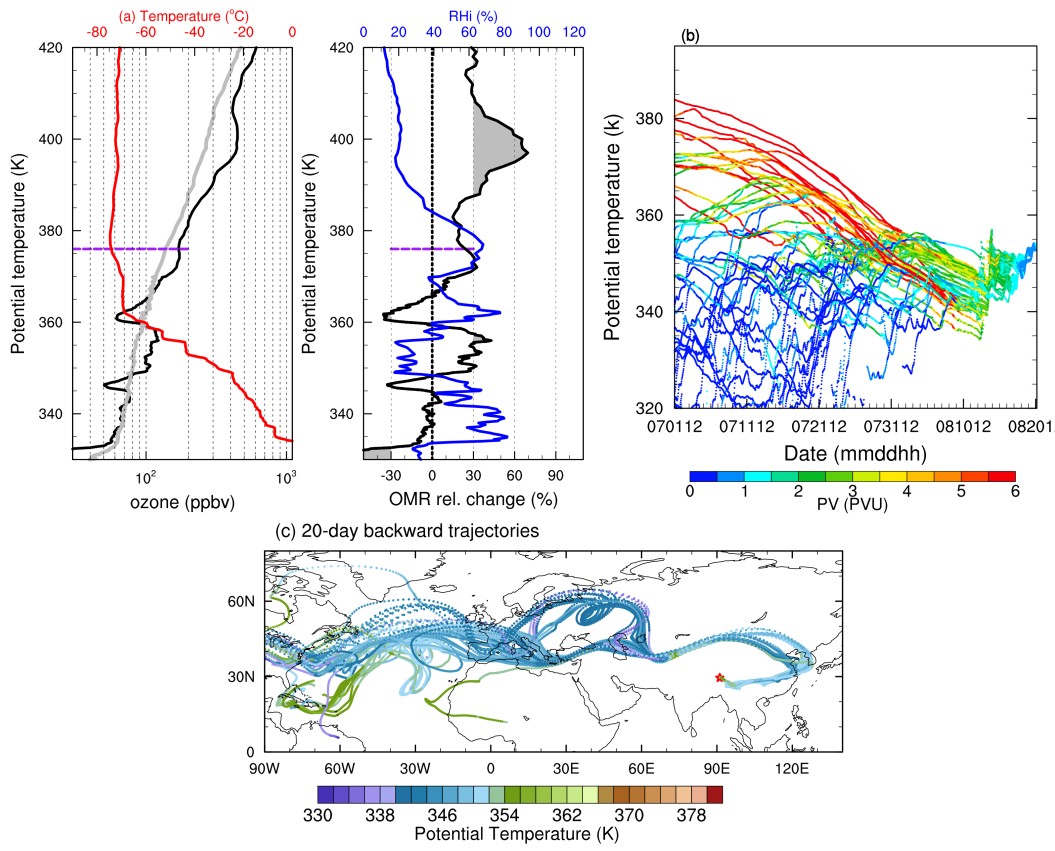

**Figure 7.** The same as Fig. 5, but on 20 August 2013.

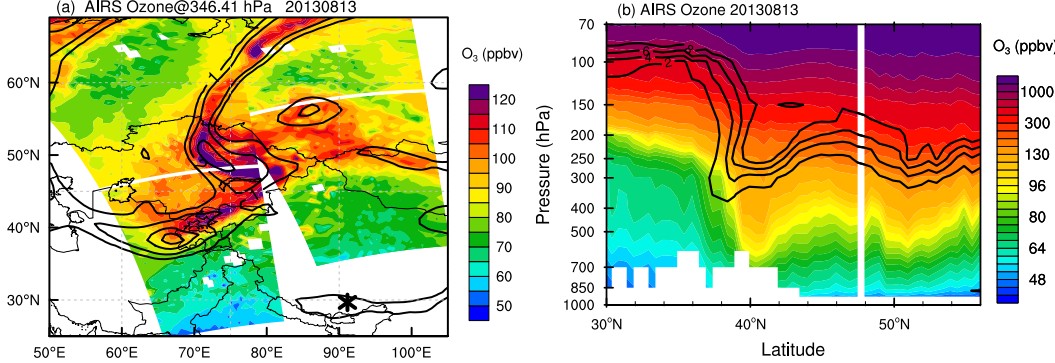

**Figure 8.** (a) Ozone volume mixing ratio for 346.41 hPa on 13 August at about 06:00 UTC from AIRS satellite. (b) Latitude−pressure cross section of ozone along 70° E with PV (2, 4, 6, and 8 PVU, black line).

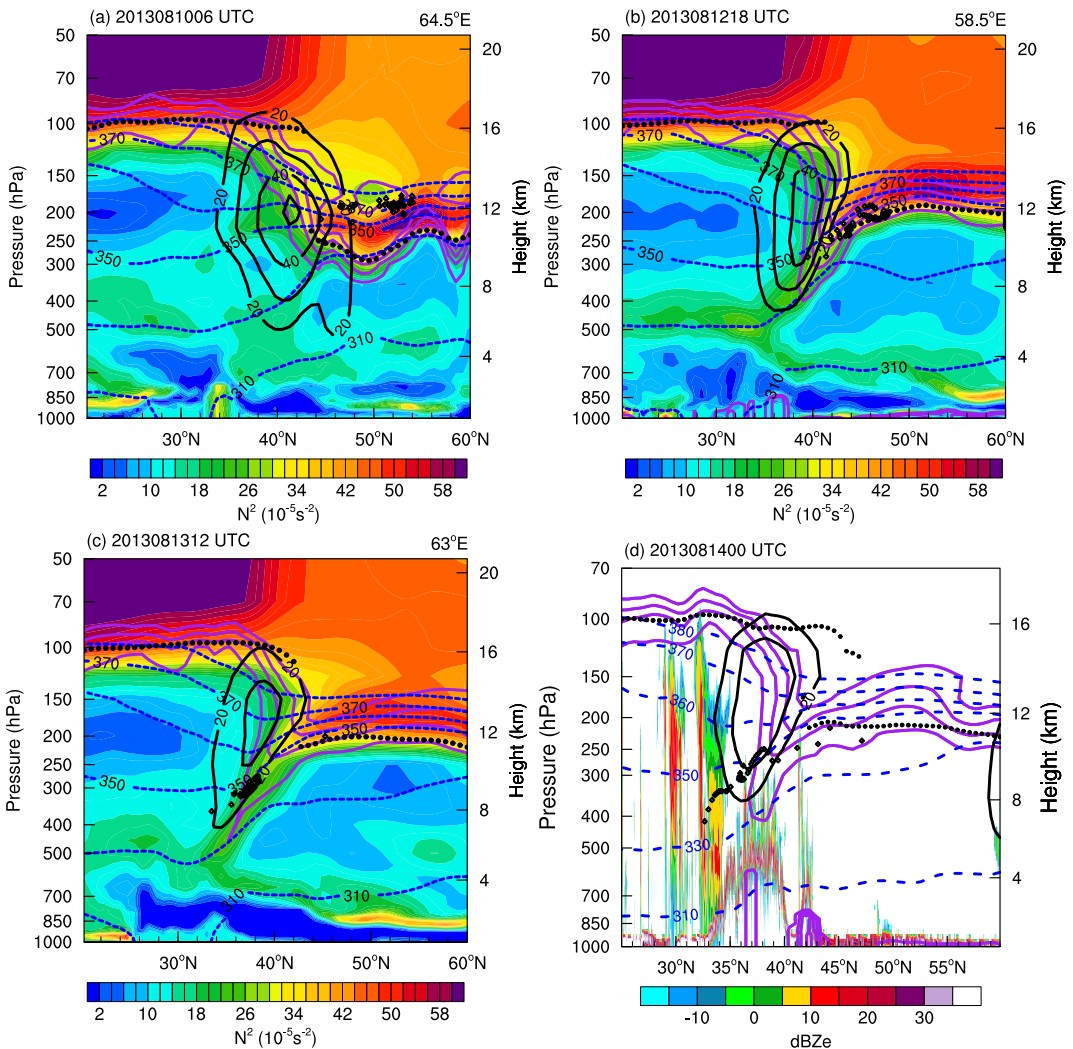

**Figure 9.** Latitude−pressure cross section of the square of the Brunt−Vaisala frequency ($N^2$) along the average longitude of the bottom air parcels in Fig. 5b on (a) 10 August at 06:00 UTC, (b) 12 August at 18:00 UTC, (c) 13 August at 12:00 UTC with potential temperature (blue dashed lines, K), PV (purple lines, PVU), tropopause (black dots, hPa), zonal wind (black lines) from ERA-Interim, and air parcels (diamonds) from CLaMS trajectory model. (d) The same as Figure 6a except for radar reflectivity (dBZe) but on 14 August at 00:00 UTC from CloudSat.