# Peer review of "High tropospheric ozone in Lhasa within the Asian summer monsoon anticyclone in 2013: influence of convective transport and stratospheric intrusions"

_Atmospheric Chemistry and Physics, 2018_

## Referee Comment (RC1) · Anonymous Referee #2 · 30 Aug 2018

This paper presents three cases of high ozone mixing ratios in the troposphere at Lhasa (29.66 N, 91.14 E) using Balloon-borne measurements. The ozone enhancement is linked to transport of polluted air from the boundary layer and downward transport from the stratosphere by intrusions. Results support with the simulation of Chemical Lagrangian Model of the Stratosphere and satellite measurements by OMI and AIRS. This paper shows important results of ozone variability over the Asian summer monsoon anticyclone region. I recommend publication after inclusion of following suggestions.

[Figure]

(1) The Introduction section is weak. The significance of the study is not well explained. A question addressed in the study should be in the interest of the larger community.

(2) Does ozone increase during few epochs have implication on upper troposphere?

(3) Present study show features from total column ozone. Ozone in the troposphere and stratosphere has different production and loss processes. During the monsoon season, lightning is one of the important agents for upper tropospheric ozone production. I suggest including a discussion on these aspects.

(4) I suggest showing vertical ozone variations using satellite observations to show stratospheric/tropospheric intrusions. It may be evident in ozone anomalies.

(5) Spatial plots of PV on potential temperature surfaces (360K or 380 K) will be useful to explain equatorward transport.

(6) Vertical cross sections from CLaMS simulations indicating stratospheric contribution for the period 18−20 August 2013 will be useful.

---

## Referee Comment (RC2) · Anonymous Referee #1 · 31 Aug 2018

Review of Manuscript acp-2018-652

**High tropospheric ozone in Lhasa within the Asian summer monsoon anticyclone 2013: influence of convective transport and stratospheric intrusions**

by Dan Li et al.

August 31, 2018

The manuscript describes promising studies in a very important region that can receive air from a number of sources. In particular, the role of stratosphere-troposphere transport is very high due to the extreme elevation of the Tibetan plateau and the presence of the subtropical jet stream. I recommend the paper for publication after some modification.

*Comments:*

P.1, line 18: Please, add a reference here. This statement is not obvious!

P. 2, line 19: I think Skerlak et al, Atmos. Chem. Phys., 14, 913-937, 2014, mention the Tibetan plateau to be a "hot spot". You should write something like "particulary active region". This a strong motivation for your measurements!!!

P. 2, line 26: "over northern India": Here (or in the following paragraph) you should mention Ohja et al., Atmos. Environ., 88, 201-211, 2014, and Atmos. Chem. Phys., 17 6743-6757, 2017.

P. 2, around line 28: I am missing some statement on the role of the subtropical jet stream (e.g., Koch et al., Int. J. Climatol. **26** (2006), 283-301; Trickl et al., Atmos. Chem. Phys. **11** (2011), 9343-9366; and references therein). This also adds to the motivation for the paper!

P. 2, line 29: Here (or below) you should add a sentence on the importance of intensifying observations in this interesting region.

P. 3, line 15: Please, specify SWOP.

P. 4, line 31: P. 4, Sec. 2.3: There is a strong need for justifying the extension of trajectory calculations to as much as 50 days!!! There are papers on the quite limited accuracy of trajectories (e.g., Stohl et al.). I think that 10 days are acceptable in the free troposphere due at least for coherent air streams. However, I have seen reasonable results in the literature times up to 20 days in certain cases.

P. 5, line 4: extremely high

P. 5, line 11: RH profiles

P. 7, line 11: Can you make conclusions about the quality of the trajectories from the results (e.g., from the coherence properties)?

Conclusions: Section 4 looks rather technical. I am missing more scientific statements in relation to the topics mentioned in the introduction. In addition, it would be advantageous to learn (e.g.) what was the idea behind the effort and what is planned. Long-term measurement would be great!

---

## Author Comment (AC2) · 30 Nov 2018

**Authors Reply to Anonymous Referee #2**

We thank the anonymous referee #2 for your useful suggestions which help us to improve the manuscript. Our detailed replies are shown below. The text with all changes in manuscript are highlighted in red color.

5

1. The Introduction section is weak. The significance of the study is not well explained. A question addressed in the study should be in the interest of the larger community.

**Reply:** The authors added more information about importance of this study in introduction. Please check the revised manuscript with tracked changes.

10

2. Does ozone increase during few epochs have implication on upper troposphere?

**Reply:** In our case studies, we find that the increasing ozone in the upper troposphere transported within the Asian summer monsoon anticyclone with a timescale of several days. An detailed statistical analysis of the vertical structure of ozone and water vapour using the complete SWOP data set over nearly 10 years is work in progress. In Li et al., 2018, the focus is to

15 understand the detailed transport pathways. A single "ozone increase episodes" influence the local chemical composition of the troposphere, however, frequently occurring "ozone increase episodes" have the potential to change the radiation balance of the atmosphere in the region of the ASM.

3.Present study show features from total column ozone. Ozone in the troposphere and stratosphere has different production and
loss processes. During the monsoon season, lightning is one of the important agents for upper tropospheric ozone production.
I suggest including a discussion on these aspects.

**Reply:** We replaced Figs. 4b–d with ERA-Interim PV on 370 K, AIRS ozone and water vapour mixing ratios at 83 hPa to show the equatorward transport process (please see Fig. 1).

- The authors added the following sentences in discussion and conclusions. "During the ASM period, deep convective clouds occurred over the South Asian subcontinent, the southern Himalayan region and the Tibetan Plateau that caused lightning activity (Qie et al., 2014). NOx produced by lighting has the potential to impact upper tropospheric ozone via photochemical reactions (Kita et al., 2002; Schumann and Huntrieser, 2007; Fadnavis et al., 2015; Gottschaldt et al., 2018). Uplifted ozone precursors and NOx produced by lighting contribute to photochemical ozone production in the upper troposphere in the ASM anticyclone. An ozone increase by 30% is found over the monsoon convection region due to NOx enhanced by lightning
- 30 according to sensitivity simulations with and without lightning (Fadnavis et al., 2015). In our case studies, ozone is increased by 90%, compared to monthly mean ozone, this value is still higher than the ozone increase due to lightning reported by Fadnavis et al. (2015). Lightning is an additional source of tropospheric ozone, which is not included in the trajectory calculations neglecting chemistry employed here. However, in addition to lightning produced  $NO_x$ , ozone precursors likely uplifted by convection are necessary. Therefore, convective uplift is the basis transport mechanism for enhanced ozone in the middle and
- 35 upper troposphere. Enhanced ozone measured over Lhasa is caused by uplift of ozone-rich air from the lower troposphere or produced by lightning, it is likely caused by a combination of both with a transport time from the planetary boundary layer to Lhasa of 20–30 days"

---

## Author Response (AR2)

**Authors Reply to Anonymous Referee #1**

**The authors appreciate the anonymous referee #1 for your useful comments which are very helpful to improve our manuscript. Please see below our responses point by point.**

**General comments**

*P.1, line 18: Please, add a reference here. This statement is not obvious!*

**Reply:** The authors rewrote the first sentence from "In the upper troposphere, ozone acts as an important greenhouse gas and, thus, plays a major role in regional energy balance and in climate change." to "In the troposphere, ozone acts as an important greenhouse gas, which has a positive radiative forcing $(0.4 \pm 0.2\, W\, m^{-2})$ through the direct or indirect greenhouse effect for the period 1750–2011. Although relatively short lived, it is therefore very important for the radiation balance of the Earth's atmosphere (Myhre et al., 2013).". Further, we added a reference.

*P. 2, line 19: I think Skerlak et al, Atmos. Chem. Phys., 14, 913-937, 2014, mention the Tibetan plateau to be a "hot spot". You should write something like "particularly active region". This is a strong motivation for your measurements!!!*

**Reply:** We rewrote this sentence from " The ASM anticyclone is also an active region for stratosphere−troposphere exchange (e.g., Škerlak et al., 2014; Garny and Randel, 2016; Fan et al., 2017). " to "The ASM anticyclone is an active region for both troposphere-to-stratosphere stratosphere-to-troposphere transport (e.g., Garny and Randel, 2016; Fan et al., 2017), particularly the Tibetan Plateau region (Škerlak et al., 2014)". And we added the following sentence "Ozone concentration in the planetary boundary layer over the Tibetan Plateau is likely affected by intense deep stratospheric intrusions (Škerlak et al., 2014)." in this paragraph.

*P. 2, line 26: "over northern India": Here (or in the following paragraph) you should mention Ojha et al., Atmos. Environ., 88, 201-211, 2014, and Atmos. Chem. Phys., 17 6743-6757, 2017.*

**Reply:** The authors added the following sentence here.

"Balloon measurements over the central Himalayas have shown that stratospheric intrusions enhanced ozone concentrations in the middle and upper troposphere (Ojha et al., 2014, 2017)."

*P. 2, around line 28: I am missing some statement on the role of the subtropical jet stream (e.g., Koch et al., Int. J. Climatol. 26 (2006), 283-301; Trickl et al., Atmos. Chem. Phys. 11 (2011), 9343-9366; and references therein). This also adds to the motivation for the paper!*

**Reply:** We added the following sentences at the end of this passage.

"The upper tropospheric subtropical jets stream occur from eastern Asia to the mid-Pacific with high frequency (Koch et al., 2006). Stratosphere-to-troposphere transport along the subtropical jet stream occurs over the Pacific Ocean. This is an important process for increasing ozone in the middle and upper troposphere in the region of the ASM (Trickl et al., 2011)."

*P. 2, line 29: Here (or below) you should add a sentence on the importance of intensifying observations in this interesting region.*

**Reply:** The authors added following sentences before P.2, line 29.

"In particular, the Tibetan Plateau is a hotspot region for the two-way exchange between the stratosphere and troposphere (Škerlak et al., 2014). However, in situ measurements over this region of chemical compositions in the upper troposphere and lower stratosphere are limited (e.g., Bian et al., 2012; Li et al., 2017). Because of the sparse in situ observations over the Tibetan Plateau, there is a need for further in situ observations in this region (e.g., balloon or super-pressure balloon measurements) to obtain new insights into transport and exchange processes in this region and for climatological survey."

*P. 3, line 15: Please, specify SWOP.*

**Reply:** We specified the text "The SWOP (sounding water vapour, ozone, and particle) experiment was conducted in Lhasa (29.66° N, 91.14° E, 3650 m above sea level (a.s.l.)) in 2010, 2013, 2016, and 2018 and Kunming (25.01° N, 102.65° E, 1889 m a.s.l.) in 2009, 2011, 2012, 2014, 2015, and 2017 by Institute of Atmospheric Physics, Chinese Academy of Sciences during

the summer monsoon period. The object of the SWOP is to collect the first long-term database of ozone, water vapour, and particle over the Tibetan Plateau from surface to lower stratosphere, and then to investigate and quantify the character of ozone and water vapour transport within the ASM anticyclone". An detailed statistical analysis of the vertical structure of ozone and water vapour using the complete SWOP data set over nearly 10 years is work in progress.

*P. 4, line 31: P. 4, Sec. 2.3: There is a strong need for justifying the extension of trajectory calculations to as much as 50 days!!! There are papers on the quite limited accuracy of trajectories (e.g., Stohl et al.). I think that 10 days are acceptable in the free troposphere due at least for coherent air streams. However, I have seen reasonable results in the literature times up to 20 days in certain cases.*

**Reply:** We agree with the referee's comment about the accuracy of trajectories. The results based on a few trajectories of air parcels may be arbitrary, but there are some reasonable expectation according to a very large number of trajectories. To study transport processes in the region of the Asian monsoon, a trajectory length between few days to few months is found in the literature (e.g., Chen et al., 2012; Bergman et al., 2013; Vogel et al., 2014; Vernier et al., 2015; Garny and Randel, 2016; Li et al., 2017). To study the convective events, a trajectory length of a few days to 20 days is sufficient. Li et al. (2017) using the Lhasa's balloon data in 2013 and the CLaMS trajectory model, have shown that the air parcels with low ozone from marine boundary layer over the western Pacific are the dominant source of low ozone in the tropopause layer in Lhasa resulting from very strong uplift (1–4 days) by typhoons and the subsequent horizontal long-range transport (4–10 days) within the Asian summer monsoon anticyclone. Thus, 20-day backward trajectories from the CLaMS model for the middle and upper troposphere within the ASM anticyclone are generally appropriate. However, to analyze more aged air masses transported in the region of the Asian monsoon anticyclone or originating in the stratosphere trajectories longer than 20 days are necessary (e.g., Chen et al., 2012; Bergman et al., 2013; Vogel et al., 2014; Garny and Randel, 2016). Therefore, we use 50-day backward trajectories in this study.

*P. 5, line 4: extremely high*
**Reply:** Changed to extreme high.

*P. 5, line 11: RH profiles*
**Reply:** Corrected.

*P. 7, line 11: Can you make conclusions about the quality of the trajectories from the results (e.g., from the coherence properties)?*
**Reply:** Thanks for the useful comment. To test the uncertainties of the trajectories, we run a set of trajectories close to the location of Lhasa and found that all these trajectories show a very similar transport pattern over the last 20 days (Fig. 1). Furthermore, we analyzed the PV from ERA-Interim reanalysis on isentropic surface and cross section of ozone from AIRS, and balloon measurements. We found that the satellite observations of PV agree with the PV along the backward trajectories confirming the CLaMS trajectory calculations.

*Conclusions: Section 4 looks rather technical. I am missing more scientific statements in relation to the topics mentioned in the introduction. In addition, it would be advantageous to learn (e.g.) what was the idea behind the effort and what is planned. Long-term measurement would be great!.*
**Reply:** We rewrote section 4 to connect the question quote in introduction. Please check section 4 in our revised manuscript.

[Figure]

**Figure 1.** The backward trajectories of air parcels within the high ozone structure (355–362.3 K) as a function of time and potential temperature on 11 August. Blue/grey/red/green/purple lines mark the backward trajectories of (i, j)/(i-$\Delta$i, j)/(i+$\Delta$i, j)/(i, j-$\Delta$j)/(i, j+$\Delta$j), (i, j) marks the location of Lhasa and $\Delta$i=$\Delta$j=1 degree.

20  *loss processes. During the monsoon season, lightning is one of the important agents for upper tropospheric ozone production. I suggest including a discussion on these aspects.*
**Reply:** We replaced Figs. 4b−d with ERA-Interim PV on 370 K, AIRS ozone and water vapour mixing ratios at 83 hPa to show the equatorward transport process (please see Fig. 1).
    The authors added the following sentences in discussion and conclusions. "During the ASM period, deep convective clouds
25  occurred over the South Asian subcontinent, the southern Himalayan region and the Tibetan Plateau that caused lightning activity (Qie et al., 2014). $NO_x$ produced by lighting has the potential to impact upper tropospheric ozone via photochemical reactions (Kita et al., 2002; Schumann and Huntrieser, 2007; Fadnavis et al., 2015; Gottschaldt et al., 2018). Uplifted ozone precursors and $NO_x$ produced by lighting contribute to photochemical ozone production in the upper troposphere in the ASM anticyclone. An ozone increase by 30% is found over the monsoon convection region due to $NO_x$ enhanced by lightning
30  according to sensitivity simulations with and without lightning (Fadnavis et al., 2015). In our case studies, ozone is increased by 90%, compared to monthly mean ozone, this value is still higher than the ozone increase due to lightning reported by Fadnavis et al. (2015). Lightning is an additional source of tropospheric ozone, which is not included in the trajectory calculations neglecting chemistry employed here. However, in addtion to lightning produced $NO_x$, ozone precursors likely uplifted by convection are necessary. Therefore, convective uplift is the basis transport mechanism for enhanced ozone in the middle and
35  upper troposphere. Enhanced ozone measured over Lhasa is caused by uplift of ozone-rich air from the lower troposphere or produced by lightning, it is likely caused by a combination of both with a transport time from the planetary boundary layer to Lhasa of 20–30 days"

[Figure]

**Figure 1.** Total column ozone (DU) as measured by OMI with the geopotential height (purple line) and PV (black line, in PVU) at 150 hPa, and the sea-level pressure (white line) of tropical cyclone on (a) 16, (e) 18, and (f) 19 August 2013 using ERA-Interim data. (b) The PV (shade, PVU) on 370 K surface on 16 August 2013. (c) Ozone and (d) water vapour mixing ratios for 83.666 hPa from AIRS on 16 August 2013 with PV at 150 hPa from ERA-Interim. The asterisk marks the location of Lhasa.

*4. I suggest showing vertical ozone variations using satellite observations to show stratospheric/tropospheric intrusions. It may be evident in ozone anomalies.*

**Reply:** Thank you for this helpful comment. The authors plot the ozone mixing ratio for 346 hPa (Fig. 2a) and the latitude−pressure cross section of ozone (Fig. 2b) based on AIRS data and add the Fig. 2 in section 3.3 as Fig. 8 in the revised manuscript.

[Figure]

**Figure 2.** Ozone volume mixing ratio for 346.41 hPa from AIRS satellite (left panel). The latitude-pressure cross section of ozone along $70°$ E with PV (2, 4, 6, and 8 PVU, black line).

*5. Spatial plots of PV on potential temperature surfaces (360 K or 380 K) will be useful to explain equatorward transport.*

**Reply:** We added PV on 370 K (please see Fig. 2b) to show equatorward transport.

*6. Vertical cross sections from CLaMS simulations indicating stratospheric contribution for the period 18–20 August 2013 will be useful.*

**Reply:** The authors revised Figs. 4c and 4d using AIRS measurement to highlight equatorward transport of dry ozone-rich air from the lower stratosphere on 16 August 2013 (please see Fig. 1). The authors also added a cross section of AIRS ozone (Fig. 2b) to explain the tropopause fold occurring on 14 August 2013. Because tropospheric ozone chemistry is not included in the CLaMS model, we think it is more useful to present AIRS measurements instead of results of a 3-dimensional CLaMS simulation.

**References**

[revised manuscript text omitted]